# Geographical and sociodemographic differences in statin dispensation after acute myocardial infarction in Sweden: a register-based prospective cohort study applying analysis of individual heterogeneity and discriminatory accuracy (AIHDA) for basic comparisons of healthcare quality

Juan Merlo [1,2] Johan Öberg [1,3] Kani Khalaf [1,3] Raquel Perez-Vicente [1] George Leckie [4]

For numbered affiliations see end of article.

**Correspondence to**
Professor Juan Merlo;
juan.merlo@med.lu.se

## ABSTRACT

**Background** In Sweden, as in many other countries, official monitoring of healthcare quality is mostly focused on geographical disparities in relation to a desirable benchmark. However, current evaluations could be improved by considering: (1) The intersection of other relevant axes of inequity like age, sex, income and migration status; and (2) The existence of individual heterogeneity around averages. Therefore, using an established quality indicator (ie, dispensation of statins after acute myocardial infarction, AMI), we valuate both geographical and sociodemographic inequalities and illustrate how the analysis of individual heterogeneity and discriminatory accuracy (AIHDA) enhances such evaluations.

**Population and methods** We applied AIHDA and calculated the area under the receiver operating characteristics curve (AUC) of regional and sociodemographic differences in the statin dispensations of 35 044 patients from 21 Swedish regions and 24 sociodemographic strata who were discharged from the hospital with an AMI diagnosis between January 2011 and December 2013. Following the Swedish National Board of Health and Welfare, we used a benchmark value of 90%.

**Results** Dispensation of stains after AMI in Sweden did not reach the desired target of 90%. Regional differences were absent/very small (AUC=0.537) while sociodemographic differences were small (AUC=0.618). Women, especially those with immigrant background and older than 65 years, have the lowest proportions of statin dispensations after AMI.

**Conclusions** As the AUC statistics are small, interventions trying to achieve the benchmark value should be universal. However, special emphasis should nevertheless be directed towards women, especially older women with immigrant backgrounds.

## STRENGTHS AND LIMITATIONS OF THIS STUDY

⇒ The applied analysis of individual heterogeneity and discriminatory accuracy (AIHDA) provides an improved mapping and quantification of geographical and sociodemographic inequalities in healthcare quality.

⇒ The quality of the data in our study is high as it is based on national registers with valid and standardised information on diagnoses, medication use and sociodemographic variables.

⇒ We can directly compare our results to those presented by public authorities, such as the Swedish National Board of Health and Welfare, since the definition of the quality indicator is the same.

⇒ The definition and categorisation of the sociodemographic variables is coarse. However, more fine categorisations would excessively increase the number of strata.

⇒ The AIHDA approach can be implemented using both traditional regression or more advanced multilevel models which provide several advantages. However, traditional regression is a valid and accessible alternative.

## INTRODUCTION

In Sweden as in many other countries, policies and laws emanating from patient needs regulate and define standards of healthcare.[1] Good healthcare is defined as safe, accessible, delivered in time, health promoting, and knowledge-based and evidence-based. In addition, to be fully efficient and effective, good healthcare must be equitable. That is, on equal terms and according to patients' needs. On this background, The

Swedish National Board of Health and Welfare (NBHW) and the Swedish Association of Local Authorities and Regions define quality indicators[2] that can be compared against specific benchmarks. These quality indicators are regularly monitored for quality of performance assessment by means of simplified hospital and regional comparisons online (https://vardenisiffror.se/), and in reports such as *Open Comparisons of Healthcare Performance*.[3] The benefits of monitoring for improving healthcare quality are many.[4] However, current comparisons could be improved for several reasons.

In spite of the current evidence on the existence of healthcare inequities conditioned by sociodemographic factors (eg, age, sex, income, country of birth), today's official monitoring is mostly focused on geographical disparities, disregarding other relevant axes of possible inequity. Monitoring both geographical and sociodemographic inequalities seem necessary in order to improve the evaluation of quality of healthcare performance.[5]

In addition, current comparisons are often based on the analysis and interpretation of differences between group averages, disregarding individual heterogeneity around such averages.[6 7] This is a fundamental but often ignored question in epidemiology and healthcare evaluation.[6–10] To understand this concept, we strongly suggest the reading of a previous example[9] describing in detail this idea in non-technical terms. When comparing group averages, such as hospital averages in quality outcomes, we also need to quantify the extent of the groups' distributions of individual values (ie, heterogeneity around averages). The size of this overlap needs be considered when evaluating differences between group averages.[9]

*Also*, the criteria for quantifying the size of the group differences in official comparisons are not clearly stated. For instance, when can we say that there are large regional differences in a quality indicator? The criterion of 'statistical significance' is not sufficient as very small and meaningless differences between group averages may nonetheless prove statistically significant if the sample is large enough. Therefore, measurable differences between group averages may exist together with considerable overlap of individual heterogeneity between groups, which may render the average differences irrelevant.[9 11] Therefore, group differences should not only be assessed by simple measures of differences between averages but as the share of the total individual variance that is at the group level. Alternatively, we should quantify the accuracy of the information provided by the group differences to classify persons according to the outcome.

The analysis of individual heterogeneity and discriminatory accuracy (AIHDA) we promote in this study, is an innovative analytical strategy that has the potential to address the above criticisms and thereby improve the quantitative evaluation of healthcare quality. AIHDA is not a new statistical technique but rather a strategy of analysis that stresses the importance of focusing on components of individual heterogeneity and discriminatory accuracy and not only on differences between group averages (as it is the traditional approach). AIHDA is proving an increasingly popular approach in public health and healthcare epidemiology.[5 9 10 12–21] When applying AIHDA, we do not consider differences between group averages and differences between patients as if they are two separate and unrelated phenomena of interest. Rather, we conceptually adopt a multilevel perspective that simultaneously considers both types of differences. From this standpoint, we evaluate sociodemographic and geographical group differences by quantifying the share of the total individual variation in patient outcomes that operates at the group level. In the case of binary patient outcomes (like most quality indicators), we can analogously evaluate the accuracy of the group level for discriminating individuals with the outcome from those without the outcome of interest. Applying AIHDA, we can evaluate the relevance of specific group differences (eg, regional differences), and even compare the relative relevance of different grouping criteria (eg, sociodemographic strata vs geographical regions).

AIHDA can be applied using traditional regression[13 15 17 22] or more advanced multilevel regression analyses (MLRA) that we denominate (M)AIHDA.[5 9 10 12 14 16 18–20 23] MLRA provides conceptual and methodological advantages as compared with traditional regression analyses.[24] For instance, conceptually both kinds of analyses consider the groups as contexts but only MLRA explicitly models the groups as a second level (ie, as a random effect using technical terminology). In MLRA the relative group differences (eg, prevalence ratios (PRs)) have the grand mean (ie, the mean of the groups' means) as reference while in traditional analyses the reference is a specific group (as in our present study). MLRA can better handle groups with few individuals via providing reliability weighted estimations. The interested reader may obtain a deeper understanding on this issue elsewhere.[24–26]

While we primarily recommended the use of multilevel models,[24] for simplified routine monitoring and online presentations, traditional regression analysis provides a valid AIHDA alternative. However, independently of the statistical technique, we want to stress that we consider the groups (eg, regions or socioeconomic strata) as contexts rather than as characteristics attached to the individuals. That is, conceptually we adopt a multilevel perspective that partitions the individual differences into within groups (level 1) and between groups (level 2).

The challenge of the present study is double. First, we aim to introduce a strategy of analysis that may discomfort the readers habituated with traditional focus on group mean differences.[7] Second, we aim to present this strategy in a correct but easy way. Our intention is to provide a simple alternative for evaluation of inequalities in routine healthcare. Still the analytical approach and the way we use some concepts are innovative which may need some extra contemplation. We provide suitable references for the readers that are interested in a deeper understanding of both AIHDA[13 15 17 22] and (M)AIHDA.[5 9 10 12 14 16 18–20 23]

Motivated by the above background, we illustrate how the AIHDA approach can be applied for public monitoring of geographical as well as sociodemographic and differences in statin treatment after hospitalisation for acute myocardial infarction (AMI), which is a well-known

process indicator specifically adopted by the Swedish NBHW.[27] We analyse both regional[4] and sociodemographic differences as previous research has shown less dispensations in women,[28] immigrants,[29] elderly people[30] and in patients with low socioeconomic position.[31] For this purpose, we analyse the statin dispensations of 35 044 patients discharged from the hospital with an AMI diagnosis between January 2011 and December 2013.

## POPULATION AND METHODS
### Databases
Our investigation is based on a record-linkage database of several registers with national coverage, including the Swedish Population Register and the Longitudinal Integration Database for Health Insurance and Labour Market Studies, administered by Statistics Sweden, as well as the National Patient Register (NPR), the Cause of Death Register and the Swedish Prescribed Drug Register (SPDR) administrated by the NBHW. The NPR records both outpatient clinic and inpatient discharges from hospital with diagnoses coded according to the International Classification of Diseases, 10th edition (ICD-10). However, it does not cover information on diagnoses in primary healthcare. The SPDR contains information on all prescription drug dispensations from Swedish pharmacies. To maintain confidentiality, the registers were linked using an arbitrary serial number assigned to each individual by Statistics Sweden instead of the Swedish unique personal identification number.

### Study population
We defined the study population using the same criteria as applied by the Swedish NBHW for evaluating the quality indicator on statin dispensation after hospital discharge for AMI.

The initial population included all 40 468 patients aged 40–80 years, who were residing in Sweden by 31 December 2010, and were discharged from hospital with an *AMI diagnosis* (ICD-10: I21 or I22) between 1 January 2011 and 31 December 2013. We followed each patient between the thirteenth and eighteenth months after discharge from the index AMI hospitalisation regarding their statin dispensation from the pharmacies. We excluded 4925 patients who died between discharge from hospital and end of follow-up, 149 patients who emigrated and 350 patients with missing information on country of birth. The final study population included 35 044 patients. See figure 1 for details. The selection criteria are in concordance with those used by the Swedish authorities to define the quality indicator. Therefore, we do not question the validity of the indicator, but just present a way of analysing it.

### Assessment of variables
We defined the *quality indicator* according to the Swedish authorities as at least 90% of patients aged 40–80 years with a hospital discharge diagnosis of AMI and as having had at least one statin dispensation between the 13-month and 18-month period after the hospital discharge.[27] We define *statin use* as any recorded dispensation of statin or a statin-combination drug (The Anatomical Therapeutic Chemical codes (ATC) C10AA or C10BA) in the SPDR during the follow-up period.

We classified the patients into the 21 *administrative regions* of Sweden according to their legal residence address and used Region Skåne as the reference region in the comparisons. Administrative regions in Sweden form autonomous political units and act as principals for healthcare.

We categorised *age* into two groups: 40–64 years and 65–80 years. *Sex* was defined as male or female according to the registered information. We categorised patients according to their *country of birth* into native (ie, born in Sweden) or not (ie, immigrant).

We obtained information on *individualised disposable family income* for the years 2000, 2005 and 2010 to compute a cumulative income measure that considers the size of the household and the consumption weight of the individuals according to Statistics Sweden. For each of the 3 years, income levels were categorised into 25 quantiles using the complete Swedish population. These groups from the respective 3 years were summed up, so that each individual received a value between 3 (always in the lowest income group) and 75 (always in the highest income group). We categorised this cumulative income measure into three groups by tertiles (low, medium or high income). Individuals with missing values on income during 2000 or 2005 were assigned the tertile values for the year 2010. No individuals had missing income data for 2010. By calculating a cumulative income measure, we obtain a more stable measurement of socioeconomic position.

Finally, we constructed a *sociodemographic multicategorical variable* consisting of all possible combinations of the categories of the four sociodemographic variables: age, sex, income and immigration status. This resulted in 24 strata.

### Patient and public involvement
Because of the nature of our study, patient and public involvement is necessary. We will communicate the results of our study to politicians and healthcare professionals, who represent the public's interest.

### Statistical and epidemiological analysis
To examine the association between geographical and sociodemographic variables and the use of statins, we calculated the crude percentage of patients who were dispensed statins in each region and in each multicategorical sociodemographic strata.

Next, we constructed three regression models following the AIHDA approach. Since the prevalence of the outcome was not extreme, we measured the associations between use of statins and the explanatory variables by PRs and their 95% CIs rather than by ORs. For this purpose, we applied Cox proportional hazards regression models with

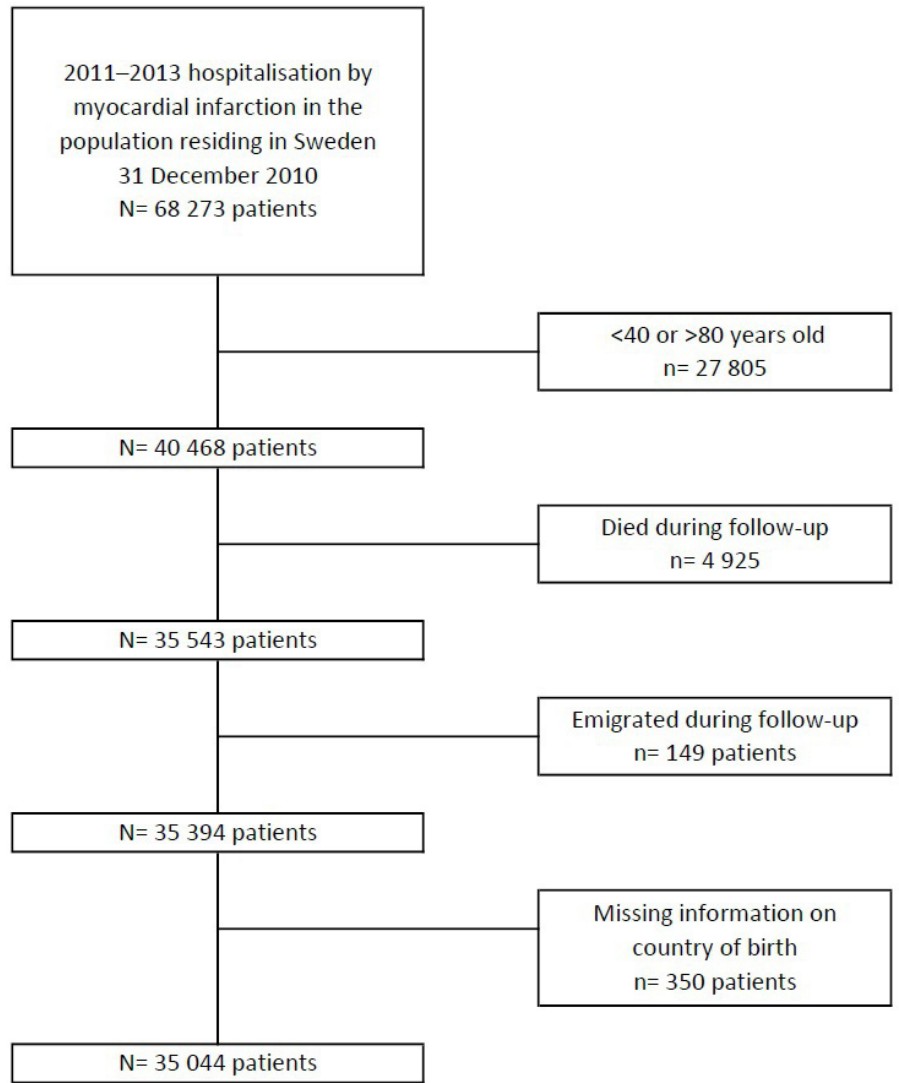

**Figure 1** Study population flow chart.

a constant follow-up time equal to 1 and report heteroskedasticity robust CIs[32] using the *vce(robust)* option in Stata (see online supplemental information S1).

*Model 1A* estimates geographical inequalities by including only the geographical variable with the 21 regions, specifying Region Skåne as reference in the comparisons.

*Model 1B* estimates sociodemographic inequalities by including the sociodemographic multicategorical variable only, specifying the strata of native men aged 40–64 years with high income as the reference category. The focus of our study was to investigate the sociodemographic multicategorical variable, but for informative reasons we performed a series of intermediate models one variable in turn from the set of variables used to construct the multicategorical variable (see online supplemental information S2).

*Model 2* enters the geographical variable and the sociodemographic multicategorical variable simultaneously. This model estimates the geographical and sociodemographic differences adjusted for each other.

In order to evaluate and compare the relevance of the sociodemographic and geographical information for understanding use of statins after AMI, we estimated the discriminatory accuracy for each model by calculating the area under the receiver operating characteristics curve (AUC) with 95% CIs. The AUC is obtained by plotting the true-positive fraction against the false-positive fraction for different thresholds of the predicted probability of statin use. The value of the AUC ranges from 0.5 to 1, with 0.5 indicating the model has no predictive accuracy and 1 representing perfect discrimination. Using the criteria proposed by Hosmer and Lemeshow,[33] we classified discriminatory accuracy as absent or very small (0.5≤AUC≤0.6), small (0.6<AUC≤0.7), large (0.7<AUC≤0.8) or very large (AUC>0.8). As explained in previous publications,[9 11] measuring geographical and sociodemographic differences to evaluate healthcare quality is analogous to performing a diagnostic test.[34] No decision maker in healthcare would want to use such a test without first knowing its discriminatory accuracy (ie, sensitivity, specificity, AUC). We do not want to

unnecessarily treat false-positive individuals or incorrectly deny treatment of false-negative patients. In our case, we want to know the accuracy of public sociodemographic and geographical comparisons for discriminating individuals with a positive result of the quality indicator (ie, use of statins) from those with a negative result (ie, no statins).

### Strategy for evaluating sociodemographic and geographical differences

Applying the AIHDA framework means that we focus on both the group averages (ie, the region and sociodemographic strata averages) and the individual heterogeneity around these averages (ie, the AUC value, which informs on the overlap in individual heterogeneity across groups).

This framework includes four steps to achieve a complete analysis of inequalities. However, the approach may be modified for different research questions.[9]

In the first step (1) We identify the quality indicator and the target or benchmark average value as described above. In the second step (2) We estimate and visualise the groups' differences using appropriate plots (eg, caterpillar plots) and in the third step (3) We quantify the size of these differences. That is, for a correct interpretation of the plots, the size of the groups' differences needs to be evaluated using a measure of discriminatory accuracy like the AUC. In traditional comparisons of group averages, the criteria for quantifying the size of the geographical and sociodemographic differences are not clearly stated. However, in the AIHDA framework we evaluate geographical and sociodemographic differences by quantifying the share of the total variation in individual outcomes that operates at the group level (ie, regions, sociodemographic strata). In the case of binary individual health outcomes, we can use the AUC as it provides information expressed in terms of discriminatory accuracy. Finally, in the fourth step (4) we proceed to the interpretation of the results. Following the study referred earlier,[9] we present a practical framework for evaluating geographical and sociodemographic differences. For this purpose, we need at least two types of information.

First, we need to know whether the average quality indicator (ie, percentage of patients on statins after AMI) has reached the predetermined target value fully (ie, ≥90% of all patients with AMI are on statins), closely (ie, 80%–89%)

or insufficiently (ie, <80%). The criteria we use to qualify the prevalence value as 'fully' is based on the definition given by the Swedish NBHW. However, the qualifications 'closely' and 'insufficiently' are ours and therefore subjective. More generally, the thresholds defining these categories will differ across quality indicators.

Second, we need information on the *size of the group differences* as assessed by the AUC, which expresses the accuracy of the differences between group averages for discriminating patients with the outcome from those without the outcome. These two types of information need to be combined for the evaluation. Table 1 illustrates these ideas by means of a simple framework with 12 scenarios that can be used to orient the interpretation of an analysis.

For this purpose, we need information about the *target indicator value*. That is, whether the average quality indicator (ie, percentage of patients on statins after AMI) has 'insufficiently', 'closely' or 'fully' reached a predetermined target *benchmark* level. We also need information on the AUC, which expresses the accuracy of the differences between group averages for discriminating patients with the outcome from those without the outcome. Combining this information, we obtain 12 different scenarios useful for the evaluation.

In the ideal scenario (scenario C) the desired target level has been fully reached in the population of patients, and the group differences are absent or very small. The conclusion would be that all groups have been treated appropriately and on equal terms. From this perspective, the quality of the healthcare is high.

In the worst scenario (scenario A) the desired target level has not been achieved in the population of patients, and the group differences are again absent. The conclusion would be that all groups have been treated on equal terms but inadequately. That is, the quality of healthcare in the whole population is low.

Observe that in both scenarios A and C interventions targeted to specific groups are not justifiable. Rather any intervention should be universal (ie, directed to the whole country). In scenario C, the reason for the intervention would be to maintain the desirable value of the quality indicator in all groups. In the scenario A, the reason would be to reach this desirable level in all the groups.

**Table 1** Framework for evaluating geographical and sociodemographic differences in a specific quality indicator

| | | Target indicator value reached | | |
|---|---|---|---|---|
| **Size of the group differences** | | **Insufficiently <80%** | **Closely 80%–89%** | **Fully ≥90%** |
| Absent/very small | 0.5≤AUC≤0.6 | A | B | C |
| Small | 0.6≤AUC<0.7 | D | E | F |
| Large | 0.7≤AUC<0.8 | G | H | I |
| Very large | 0.8≤AUC | J | K | L |

AUC, area under the receiver operating characteristics curve.

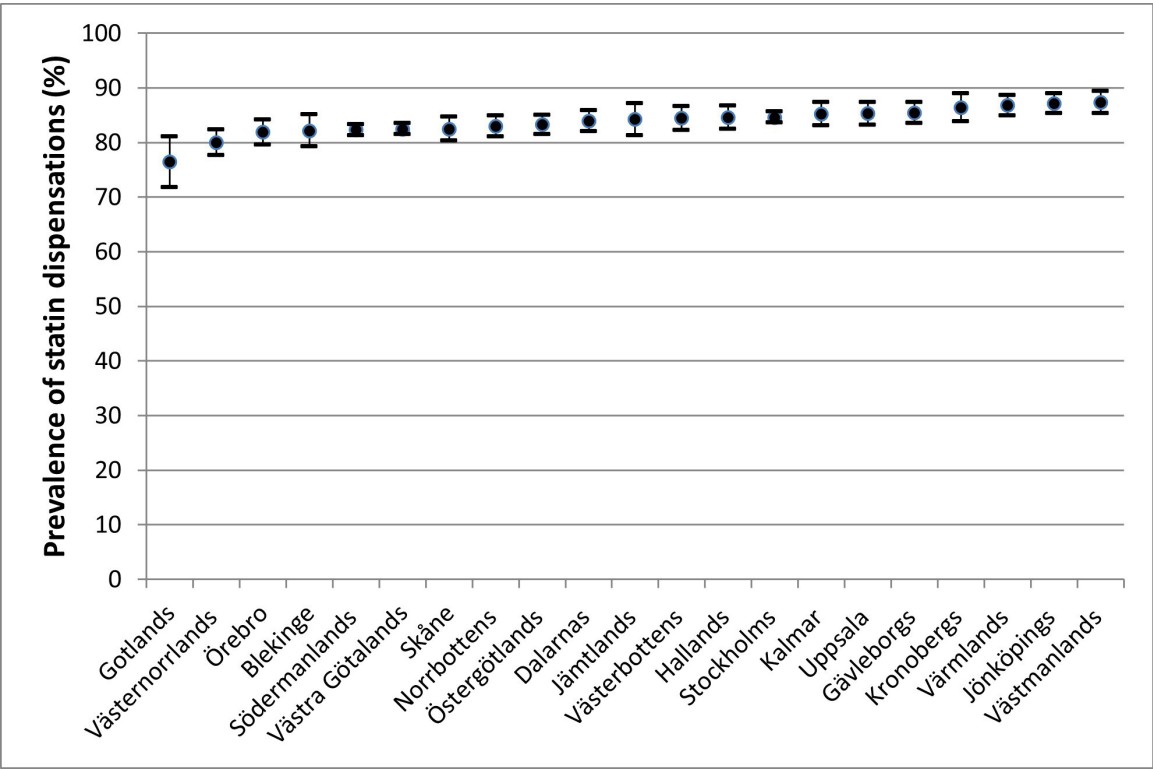

**Figure 2** Crude prevalence (P) and 95% CIs for use of statins 12–18 months after hospital discharge with an acute myocardial infarction (AMI) diagnosis in patients aged 40–80 years in the period 2011–2013, across the 21 administrative regions of Sweden.

The interpretation of the scenarios in the lowest corners of the table (J and L) is conditioned by the very large size of the group differences. For example, in scenario L even though the population as a whole has achieved the target level, some groups may not have done so. In contrast, in scenario J, some groups may have achieved the target level even though the population as a whole has not achieved it. In these scenarios (J and L) targeted group interventions are justified.

The framework we propose fits well with the concept of *proportionate universalism* for resource allocation in public health.[35 36] That is, healthcare actions must be universal, not targeted, but with a scale and intensity that is proportionate to the level of disadvantage. The AIHDA approach we apply in this study can be used to inform decisions regarding the appropriate scale and intensity for a given sociodemographic or geographical context.

## RESULTS

The overall prevalence of statin dispensations in patients with AMI 12–18 months after discharge was 83.7% (29 231/35 044). Therefore, the benchmark value of 90% was *closely* but not fully met on a national level according to the framework in table 1.

Figure 2 and table 2 illustrate the prevalence of statin dispensations in each administrative region in Sweden. Västmanland has the highest prevalence of statin use at 87.4% while Gotland has the lowest prevalence at 76.4%. None of the regions reach the benchmark value of 90%.

The relative differences when comparing with the region of Skåne (Model 1A) were tiny and table 2 illustrates that only four regions (ie, Gotland, Västernorrland, Stockholm and Västra Götaland) presented a slightly but conclusively lower PR than Skåne. No other region showed a conclusively higher PR than Skåne. The PR adjusted for the multicategorical sociodemographic variable (Model 2) were very similar to the unadjusted PR (Model 1A), indicating that the sociodemographic composition of the regions did not confound the geographical differences.

Figure 3 and table 3 illustrate the crude prevalence of statin dispensations for each sociodemographic stratum, with the highest prevalence (ie, 89.5%) in the reference strata of native men with high income aged 40–64 years, followed by the strata of native men with middle income aged 40–64 years (ie, 88.8%). We found the lowest prevalence of statin use in the stratum of immigrant women with high income aged 65–80 years (ie, 70.3%), closely followed by immigrant women with low income aged 65–80 years (ie, 74.1%). Only the reference sociodemographic stratum almost grasped the target value of 90% (ie, 89.5%). None of the female strata were close to reaching the target. In contrast, the male strata were often much closer to reaching the target. Table 3 also presents the unadjusted PR (Model 1B), and the PR adjusted for the geographical variable (Model 2), with the strata of high-income native men aged 40–64 years as the reference stratum. As many as 19 sociodemographic strata show conclusively smaller PRs than the reference.

**Table 2** Number of patients using statins (n), number of individuals (N), crude prevalence (P) and 95% CIs, unadjusted prevalence ratios (PR), and prevalence ratios adjusted for sociodemographic variables (PRa) for use of statins 12–18 months after hospital discharge with an acute myocardial infarction (AMI) diagnosis in patients aged 40–80 years from the 21 Swedish geographical administrative regions in the period 2011–2013

| | n | N | P (95% CI) % | Model 1A<br>PR (95% CI) | Model 2<br>PRa (95% CI) |
|---|---|---|---|---|---|
| Sweden | 29 321 | 35 044 | 83.67 (83.28 to 84.06) | — | — |
| Administrative regions | | | | | |
| Gotland | 246 | 322 | 76.40 (71.76 to 81.04) | 0.90 (0.85 to 0.96) | 0.91 (0.85 to 0.97) |
| Västernorrland | 922 | 1153 | 79.97 (77.65 to 82.28) | 0.95 (0.92 to 0.98) | 0.94 (0.92 to 0.97) |
| Örebro | 886 | 1082 | 81.89 (79.59 to 84.18) | 0.97 (0.94 to 1.00) | 0.97 (0.94 to 1.00) |
| Blekinge | 525 | 639 | 82.16 (79.19 to 85.13) | 0.97 (0.93 to 1.01) | 0.98 (0.94 to 1.02) |
| Stockholm | 4477 | 5440 | 82.30 (81.28 to 83.31) | 0.97 (0.96 to 0.99) | 0.97 (0.95 to 0.98) |
| Västra Götaland | 4391 | 5326 | 82.44 (81.42 to 83.47) | 0.97 (0.96 to 0.99) | 0.97 (0.96 to 0.99) |
| Uppsala | 960 | 1164 | 82.47 (80.29 to 84.66) | 0.97 (0.95 to 1.00) | 0.97 (0.94 to 1.00) |
| Norrbotten | 1215 | 1465 | 82.94 (81.01 to 84.86) | 0.98 (0.95 to 1.01) | 0.98 (0.95 to 1.00) |
| Östergötland | 1400 | 1682 | 83.23 (81.45 to 85.02) | 0.98 (0.96 to 1.01) | 0.98 (0.96 to 1.01) |
| Dalarna | 1212 | 1444 | 83.93 (82.04 to 85.83) | 0.99 (0.97 to 1.02) | 0.99 (0.97 to 1.02) |
| Jämtland | 501 | 595 | 84.20 (81.27 to 87.13) | 1.00 (0.96 to 1.03) | 1.00 (0.96 to 1.03) |
| Västerbotten | 893 | 1058 | 84.40 (82.22 to 86.59) | 1.00 (0.97 to 1.03) | 0.99 (0.97 to 1.02) |
| Halland | 940 | 1112 | 84.53 (82.41 to 86.66) | 1.00 (0.97 to 1.03) | 1.00 (0.97 to 1.02) |
| Skåne | 4085 | 4828 | 84.61 (83.59 to 85.63) | Reference | Reference |
| Kalmar | 898 | 1054 | 85.20 (83.06 to 87.34) | 1.01 (0.98 to 1.04) | 1.01 (0.98 to 1.04) |
| Södermanland | 973 | 1141 | 85.28 (83.22 to 87.33) | 1.01 (0.98 to 1.04) | 1.01 (0.98 to 1.04) |
| Gävleborg | 1126 | 1318 | 85.43 (83.53 to 87.34) | 1.01 (0.98 to 1.04) | 1.01 (0.99 to 1.04) |
| Kronoberg | 592 | 685 | 86.42 (83.86 to 88.99) | 1.02 (0.99 to 1.05) | 1.02 (0.99 to 1.05) |
| Värmland | 1076 | 1240 | 86.77 (84.89 to 88.66) | 1.03 (1.00 to 1.05) | 1.03 (1.00 to 1.05) |
| Jönköping | 1126 | 1292 | 87.15 (85.33 to 88.98) | 1.03 (1.01 to 1.06) | 1.03 (1.00 to 1.05) |
| Västmanland | 877 | 1004 | 87.35 (85.29 to 89.41) | 1.03 (1.01 to 1.06) | 1.04 (1.01 to 1.06) |
| AUC | | | | 0.537 | 0.618 |

AUC, area under the receiver operating characteristics curve.

The adjusted PRs were very similar to the unadjusted PRs indicating that geographical factors did not confound the sociodemographic differences.

Model 1A, which includes information about administrative regions, presented an AUC value of 0.537. This AUC value indicates that the geographical differences were *absent/very small,* and that region provides very limited information about statin dispensations. Model 1B, which includes information on the sociodemographic strata, showed an AUC of 0.609 which indicates that the sociodemographic differences were *small*. The sociodemographic AUC was 0.072 units higher than the AUC of the geographical model 1A. Model 2, which includes both geographical and sociodemographic information, presented an AUC value of 0.618, just 0.008 higher than that presented by model 1B. In the context of our analysis, these findings indicate that sociodemographic information is relatively more relevant than geographical information. In summary, the target indicator value was *closely* reached in Sweden as a whole. The geographical analysis indicated *absent/very small* differences, which corresponds with scenario B in table 1. The sociodemographic analysis indicated *small* differences, which corresponds with the scenario E in table 1.

In relation to the sociodemographic differences, the complementary analysis (S1) showed that the AUC was very small when we fitted separate models which included only one variable at a time. The highest AUC was for the variable sex (AUC=0.579) which was 0.029 units lower than the AUC for all variables together (AUC=0.608). We also observed a very small interaction of effects in the model with the multicategorical variable as when comparing with a model with all the single variables together, the AUC of the model with the multicategorical variable only increases 0.001 units.

## DISCUSSION

We evaluated regional and sociodemographic inequalities in statin treatment after AMI. When doing so,

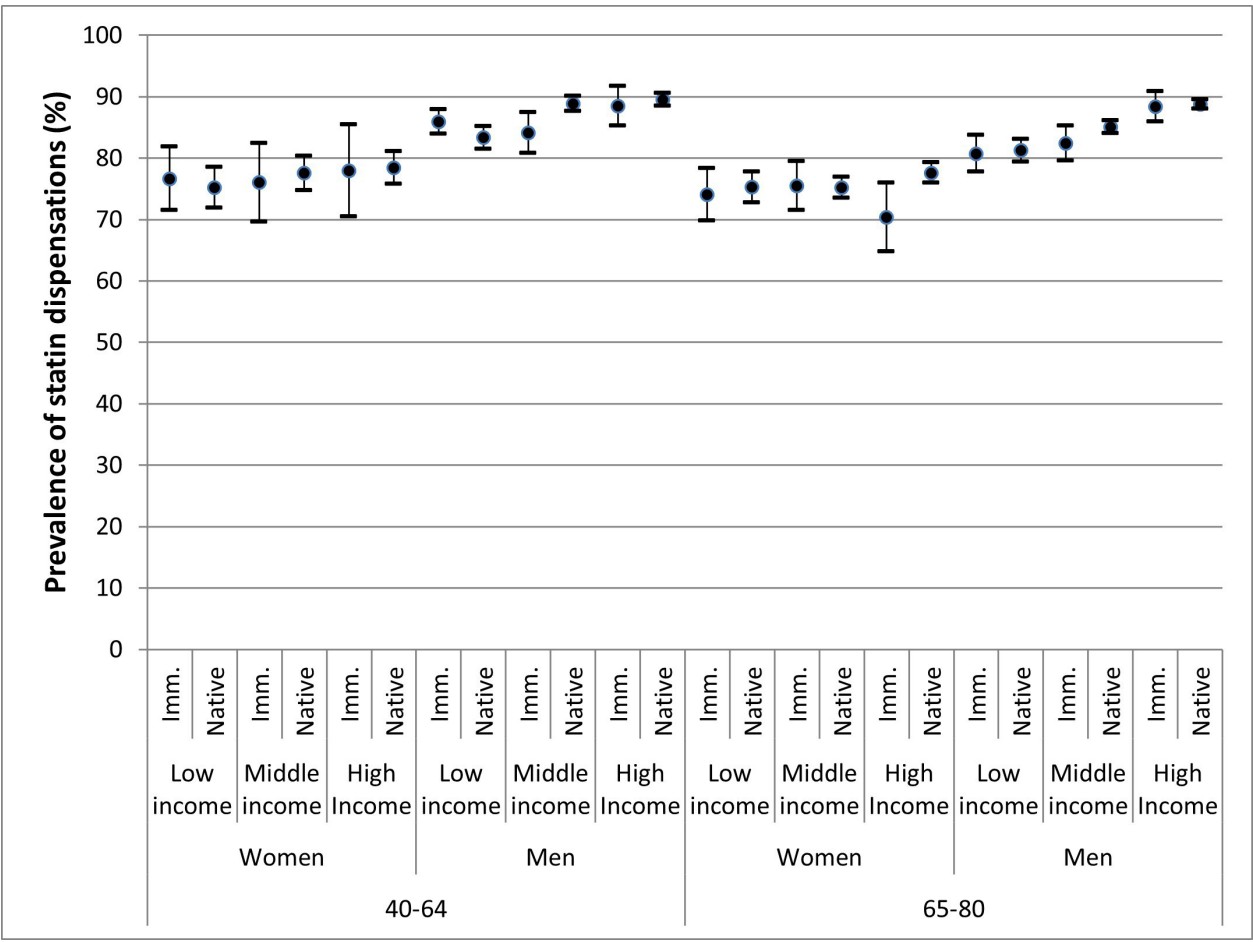

**Figure 3** Crude prevalence (P) and 95% CIs for use of statins 12–18 months after hospital discharge with an acute myocardial infarction (AMI) diagnosis in patients aged 40–80 years in the period 2011–2013, across the 24 sociodemographic groups.

we also aimed to demonstrate how AIHDA provides enhanced information for evaluating healthcare inequalities.[9 37] While traditional public comparisons focus on differences between group averages, AIHDA considers both differences between group averages and the individual heterogeneity around those averages as expressed by their discriminatory accuracy. AIHDA informs on the share of the total individual variation in patient outcomes that operates at the group level. That is, to which degree the group-level information is able to discriminate with accuracy patients with, from those without the indicator under study.[9] The idea of discriminatory accuracy is fundamental in many fields of epidemiology not only when evaluating statistical model prediction,[38] but also in the evaluation of diagnostic and prognostic tests,[34] as well as risk factors in public health[6] and in the study of geographical[37] and socioeconomic[19] determinates of health. It is also a cornerstone in current precision medicine.[39] Nevertheless, these concepts need wider adoption and we refer elsewhere for a longer explanation.[6 7 9 10]

The overall prevalence of statin treatment in our study population was 83.7% which, while lower, was judged high enough to have 'closely' reached the target benchmark value of 90%. We observed regional and sociodemographic differences in the prevalence of statin treatment after AMI. However, the low AUC values indicated that the differences were 'absent/very small' (AUC=0.537) for the administrative regions and 'small' (AUC=0.609) for the sociodemographic groups. Combining both regional and sociodemographic information did not substantially increase the AUC of the sociodemographic information alone (AUC=0.618). That is, the sociodemographic differences seem to play a modest but more relevant role than the regional differences for understanding inequalities in statin treatment after AMI.

Our analysis also shows that women in all strata have a lower prevalence of statin dispensations after AMI than men, which is in line with previous research indicating the existence of gender inequalities.[28] However, those inequalities need be interpreted in the light of the small discriminatory accuracy of the sociodemographic strata. In fact, while the prevalence of using statins is higher among men, the fact that patients with AMI are disproportionately male means that majority of non-statin users are nevertheless also men.

According to the framework presented in table 1, the sociodemographic differences would be placed in scenario E. That is, the target value was closely but not fully reached and there were small differences between the sociodemographic strata. Accordingly, any strategy

Table 3 Number of patients using statins (n), number of individuals (N), crude prevalence (P) and 95% CIs, unadjusted prevalence ratios (PR), and prevalence ratios adjusted for administrative regions (PRa) for use of statins 12–18 months after hospital discharge with an acute myocardial infarction (AMI) diagnosis in patients aged 40–80 years from 24 sociodemographic strata in the period 2011–2013

| | n | N | P (95% CI) % | Model 1B<br>PR (95% CI) | Model 2<br>PRa (95% CI) |
|---|---|---|---|---|---|
| Sweden | 29 321 | 35 044 | 83.67 (83.28 to 84.06) | — | — |
| Women | | | | | |
| 65–80, high income, immigrant | 180 | 256 | 70.31 (64.72 to 75.91) | 0.79 (0.72 to 0.85) | 0.79 (0.73 to 0.85) |
| 65–80, low income, immigrant | 300 | 405 | 74.07 (69.81 to 78.34) | 0.83 (0.78 to 0.88) | 0.83 (0.78 to 0.88) |
| 65–80, middle income, native | 1936 | 2575 | 75.18 (73.52 to 76.85) | 0.84 (0.82 to 0.86) | 0.84 (0.82 to 0.86) |
| 40–64, low income, native | 479 | 637 | 75.20 (71.84 to 78.55) | 0.84 (0.80 to 0.88) | 0.84 (0.80 to 0.88) |
| 65–80, low income, native | 852 | 1132 | 75.27 (72.75 to 77.78) | 0.84 (0.81 to 0.87) | 0.84 (0.81 to 0.87) |
| 65–80, middle income, immigrant | 338 | 448 | 75.45 (71.46 to 79.43) | 0.84 (0.80 to 0.89) | 0.84 (0.80 to 0.89) |
| 40–64, middle income, immigrant | 130 | 171 | 76.02 (69.62 to 82.42) | 0.85 (0.78 to 0.92) | 0.85 (0.78 to 0.93) |
| 40–64, low income, immigrant | 197 | 257 | 76.65 (71.48 to 81.83) | 0.86 (0.80 to 0.92) | 0.86 (0.80 to 0.92) |
| 40–64, middle income, native | 656 | 846 | 77.54 (74.73 to 80.35) | 0.87 (0.83 to 0.90) | 0.86 (0.83 to 0.90) |
| 65–80, high income, native | 1838 | 2369 | 77.59 (75.91 to 79.26) | 0.87 (0.85 to 0.89) | 0.87 (0.85 to 0.89) |
| 40–64, high income, immigrant | 92 | 118 | 77.97 (70.49 to 85.44) | 0.87 (0.79 to 0.96) | 0.87 (0.79 to 0.96) |
| 40–64, high income, native | 709 | 904 | 78.43 (75.75 to 81.11) | 0.88 (0.85 to 0.91) | 0.87 (0.84 to 0.91) |
| Men | | | | | |
| 65–80, low income, immigrant | 548 | 679 | 80.71 (77.74 to 83.67) | 0.90 (0.87 to 0.94) | 0.90 (0.87 to 0.94) |
| 65–80, low income, native | 1409 | 1735 | 81.21 (79.37 to 83.05) | 0.91 (0.88 to 0.93) | 0.91 (0.88 to 0.93) |
| 65–80, middle income, immigrant | 565 | 686 | 82.36 (79.51 to 85.21) | 0.92 (0.89 to 0.95) | 0.92 (0.89 to 0.95) |
| 40–64, low income, native | 1283 | 1540 | 83.31 (81.45 to 85.17) | 0.93 (0.91 to 0.95) | 0.93 (0.91 to 0.95) |
| 40–64, middle income, immigrant | 397 | 472 | 84.11 (80.81 to 87.41) | 0.94 (0.90 to 0.98) | 0.94 (0.90 to 0.98) |
| 65–80, middle income, native | 3791 | 4458 | 85.04 (85.04 to 85.04) | 0.95 (0.93 to 0.97) | 0.95 (0.93 to 0.96) |
| 40–64, low income, immigrant | 1006 | 1171 | 85.91 (83.92 to 87.90) | 0.96 (0.94 to 0.98) | 0.96 (0.94 to 0.99) |
| 65–80, high income, immigrant | 570 | 645 | 88.37 (88.37 to 88.37) | 0.99 (0.96 to 1.02) | 0.99 (0.96 to 1.02) |
| 40–64, high income, immigrant | 337 | 381 | 88.45 (85.24 to 91.66) | 0.99 (0.95 to 1.03) | 0.99 (0.95 to 1.03) |
| 65–80, high income, native | 6303 | 7101 | 88.76 (88.76 to 88.76) | 0.99 (0.98 to 1.01) | 0.99 (0.98 to 1.00) |
| 40–64, middle income, native | 2240 | 2522 | 88.82 (87.59 to 90.05) | 0.99 (0.97 to 1.01) | 0.99 (0.97 to 1.01) |
| 40–64, high income, native | 3165 | 3536 | 89.51 (88.50 to 90.52) | Reference | Reference |
| AUC | | | | 0.609 | 0.618 |

AUC, area under the receiver operating characteristics curve.

aimed to improve the outcome of the quality indicator should be universal but proportionately more intense among sociodemographic groups with increased risk not using statins. Analogously, the differences between administrative regions would be placed in the scenario B. In this case we should recommend universal interventions without targeting any specific regions.

### Strengths and limitations

In addition to an improved and more nuanced analysis, AIHDA conveys further strengths as compared with traditional studies exclusively based on differences between group averages. It is known that epidemiological studies comparing group averages may stigmatise the individuals belonging to groups with 'bad' average values and create *false expectations* for the individuals belonging to groups with 'good' average values. This situation is true for both traditional and AIHDA studies and it is an inevitable side effect in epidemiology. Nevertheless, the perils of stigmatisation and false expectations are often ethically justifiable because the aim of the epidemiological analyses is to improve the health and heathcare of the individuals. However, knowing the discriminatory accuracy of the group comparisons allows us to decide whether pointing out specific population groups is justifiable or not. If the discriminatory accuracy is high, pointing out specific population groups is justifiable. However,

if the discriminatory accuracy is low, the traditional emphasis on differences between group averages should be avoided. For instance, our results show that, because the AUC value is low, differences between administrative regions are less relevant for understanding individual statin dispensations. Therefore, simplistic interpretation based on visual analyses of figures like the caterpillar plot in our study (figure 2) should be sidestepped.

By overlooking the discriminatory accuracy of the group-averages, traditional studies may simultaneously lead to both undertreatment and overtreatment if a possible intervention is based on group information only. If the discriminatory accuracy is low, many individuals taking statins but belonging to a group with a 'bad' average (ie, false-positives cases) could be exposed to the intervention while many individuals not taking statins but belonging to a group with a 'good' average (ie, false-negative cases) would not receive any intervention. For instance, using the information in table 2, we can see that the region of Gotland has the worst average value (ie, 76.4%), meaning that 76 of the 322 patients were not dispensed statins. However, in the region of Västmanland with the best average value (ie, 87.4 %), this figure was 127 of the 1004 patients since Västmanland has a larger population. That is, the 'best' region has 1.7 (=127/76) times more patients lacking statin treatment that the 'worst' region. It is correct that the region of Västmanland has a higher proportion of treated patients but from a national perspective the burden of untreated patients is also higher in the region of Västmanland. Therefore, if the discriminatory accuracy is low, any potential intervention to increase statin treatment after AMI needs to be universal (ie, on the whole country) and directed to all patients with AMI no matter the region they belong to.

Similarly, in table 3 we can estimate that the number of male patients with AMI without statins (n=3312) was much higher than the number of female patients with AMI without statins (n=2411) even though women consistently have a lower prevalence and a lower PR of statin dispensation than men. This reflects that while the prevalence of using statins is higher among men, the fact that patients with AMI are disproportionately male means that majority of non-statin users are nevertheless also men.

One further strength of this study is the quality of the data since all information is based on national registers where the information is recorded in a highly standardised way for both ICD and ATC codes. The validity of the AMI diagnosis as recorded in the NPR is high,[40] and the sociodemographic information is recorded by Statistics Sweden with a long-standing experience in the field (https://www.scb.se/en/About-us/main-activity/quality-work/). Moreover, we can directly compare our results to those presented by the NBHW since the definition of the quality indicator is the same.

One limitation of our study is that we excluded the 5424 individuals who died, emigrated or had missing information on country of birth which corresponds to approximately 13% of the final study population, and

it cannot be ruled out that this exclusion introduces a selection bias. However, we do not think that this limitation alters our conclusions as the selection criteria are in concordance with those used by the Swedish authorities. Another limitation in this study could be the use of household income as a proxy for socioeconomic position, which does not capture other financial assets, nor necessarily higher education.

The definition of country of birth within this study could be elaborated with more refined categories, other than native and immigrant. Immigrant is a broad concept which includes, among other factors, cultural, economic and educational heritages from another countries. However, more categories of country of birth would result in a larger number of multicategorical strata with a smaller number of within-strata observations. Similarly, there is a possible underestimation of the impact of age since we only use two categories in our analyses. We chose two categories to keep the analysis as simple as possible and also because in the case of the quality indicator we analyse we think it is sufficient to include the age of retirement (ie, 65 years for most patients at the time of the study) as the cut-off. It is known that this division is relevant for understanding socioeconomic differences in healthcare utilisation.[41] However, for other quality indicators it may be better to provide more age classes.

We use the AUC as a measure of discriminatory accuracy and in some situations when, for example, false negatives are more likely to occur than false positives because of information bias, assuming the same weight for both false negatives and false positives might give misleading AUC values. However, this is not a problem in biomedical applications in general and in our study in particular as positive or negative events are easy to identify (a patient uses statins or not) and the information is, less probably, differentially biased. The AUC value may also depend on the size of the group being compared. For instance, a stratum may have a high predicted probability of using statins so all individuals in the stratum are classified as 'positive' but if the stratum is very large the number of individuals in the stratum who do not use statins (ie, false positive) may be large and the AUC reduced. This could be prevented by weighting the strata by the inverse of the number of individuals or using other measures of discriminatory accuracy like those based on components of variance.[9] However, in our case the unweighted results are relevant per se. A low unweighted AUC would suggest that any potential intervention to increase the use of statins should not be targeted to specific strata with a low predicted probability of statin use but be universal and directed to all the strata since there are many patients in the strata with high predicted probabilities who do not use statins. For instance, in our study 37% (2146/5723) of the patients who did not use statins reside in the 10 counties with the highest predicted probability of using statins, and this value was 58% (3312/5723) for the 12 sociodemographic strata with the highest predicted probability of using statins.

## Implications and conclusions

Dispensations of statins after AMI in Sweden are, according to the national quality indicator, closely but not fully met in relation to the benchmark value of 90%. However, neither regional nor sociodemographic differences in group averages seem to be major determinants of dispensation of statins. Therefore, any intervention aiming to reach the benchmark value should be universal. Sociodemographic differences nevertheless appeared more relevant than geographical differences. However, our results suggest that in spite of the lower prevalence of statin prescription in some groups (eg, women, especially to those with immigrant background and older than 65 years), the exclusive targeting of interventions to those groups would be unsuitable as many patients with AMI lacking statin prescription are in the groups with higher prevalence of statin prescription. In other words, we cannot base decisions on stratum prevalence alone, we also need to consider the size of the strata.

In conclusion, as we illustrate in our study, AIHDA is not complicated to implement, and it provides more nuanced information than traditional comparisons of quality of healthcare performance based only on group averages. We therefore propose the application of the AIHDA framework for the quantitative assessment of healthcare quality by geographical and sociodemographic comparisons of quality indicators.

**Author affiliations**
[1]Unit for social epidemiology, Department of Clinical Sciences, Faculty of Medicine, Lund University, Malmö, Sweden
[2]Centre for Primary Health Care Research, Region Skåne, Malmö, Sweden
[3]Department of Health and Medical Care Management, Region Skåne, Malmö, Sweden
[4]Centre for Multilevel Modelling, University of Bristol, Bristol, UK

**Contributors** JM is the guarantor of the study and accepts full responsibility for the finished work and/or the conduct of the study, had access to the data, and controlled the decision to publish. JM initiated of the study and acquired the data. JM and JÖ wrote the original manuscript. RP-V and JÖ performed the analyses in coordination with JM, KK and GL. All the authors have contributed to the design of the study, JM and GL developed the methodology and directed the analyses. All authors have contributed to the interpretation of the results and drafting of the manuscript. All authors revised the last version of the manuscript.

**Funding** The Swedish Research Council (Vetenskapsrådet) supported this work through the project 'Multilevel Analyses of Individual Heterogeneity: innovative concepts and methodological approaches in Public Health and Social Epidemiology': (#2017-01321, PI: JM).Leckie was funded by a UK Economic and Social Research Council (ESRC) grant ES/X011313/1.

**Competing interests** None declared.

**Patient and public involvement** Patients and/or the public were involved in the design, or conduct, or reporting, or dissemination plans of this research. Refer to the Methods section for further details.

**Patient consent for publication** Not applicable.

**Ethics approval** The Regional Ethics Review Board in Southern Sweden and the data safety committees from the NBHW and from Statistics Sweden approved the construction of the database. Regional ethics committee Lund: ID: 2014/856. Data safety committees from the National Board of Health and Welfare: ID 9542/2015. Statistics Sweden: ID 231424/878144-5.

**Provenance and peer review** Not commissioned; externally peer reviewed.

**Data availability statement** Data are available upon reasonable request.

**ORCID iDs**
Juan Merlo http://orcid.org/0000-0001-8379-9708
Johan Öberg http://orcid.org/0000-0002-0701-5155
Kani Khalaf http://orcid.org/0000-0002-6749-7037
Raquel Perez-Vicente http://orcid.org/0000-0002-6273-1656
George Leckie http://orcid.org/0000-0003-1486-745X

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
