## [Reviewer comments · BMJ Open]

ARTICLE DETAILS

TITLE (PROVISIONAL)	Geographic and sociodemographic differences in statin dispensation after acute myocardial infarction in Sweden – A register based prospective cohort study applying analysis of individual heterogeneity and discriminatory accuracy (AIHDA) for basic comparisons of healthcare quality.
AUTHORS	Merlo, Juan; Öberg, Johan; Khalaf, Kani; Perez-Vicente, Raquel; Leckie, George

VERSION 1 – REVIEW

REVIEWER	Yin, Lu Chinese Academy of Medical Sciences and Peking Union Medical College Fuwai Hospital, Medical Research & Biometrics Center
REVIEW RETURNED	14-Jun-2022

GENERAL COMMENTS	Even with a large sample size, based on the poor novelty and a potentially wrong design, the study has low power to be published.
---

REVIEWER	Miglio, Rossella University of Bologna, Statistical Sciences
REVIEW RETURNED	21-Jun-2022

GENERAL COMMENTS	This work is aimed to evaluate both geographical and sociodemographic differences in the statin treatment after AMI in Swedish regions using an AIHDA approach. The method is simple and can help to understand the effect of the studied variables even if some extension should be considered in my opinion as reported below. The analysis first concentrates on geographical heterogeneity and then on sociodemographic influence on the distribution of the prevalence rate of statin dispensation. The authors construct multicategorical sociodemographic strata that basically correspond to the interactions between four qualitative sociodemographic variables: age, sex, income and immigration status. Age was considered only using two classes, this could be not effective to evaluate the age effect as reported in the limitation of the study. Moreover working in this way it is not possible to evaluate separate effects of each variable and to understand if some of them contribute more to the final results and if there is a real interaction between all this variables. The model 2 is complex and it is possible that it doesn't predict better than a model with single variables. I suggest the authors to consider also alternative models with simple separate effect of each variable even if they are not
--

	interested on a mean effect it is important to verify that the proposed model is effectively better. In the method paragraph they declare to use robust confidence intervals but it is not specified how they were derived. Moreover it is reported: “Model 1B estimates sociodemographic inequalities by including the sociodemographic multicategorical variable only, specifying the strata of native 40-64 years old men as the reference category.” I think that something is missing here, we have no information on income (middle income I suppose). Why it was not considered as an alternative a Poisson regression model (multilevel)? Considering the results the discriminatory power is not so high and this is a limit of the model on one side, probably other individual variables could improve the classification performance of the model, on the other side this have positive implication from the public health perspective. It is interesting the interpretation offered in the framework reported in box1.
--	---

REVIEWER	Christensen, Daniel Telethon Kids Institute
REVIEW RETURNED	10-Oct-2022

GENERAL COMMENTS	The paper uses a novel statistical approach to assess the question of whether statin dispensation targets are met across regions and sociodemographic strata of Sweden. This paper addresses a clearly policy-relevant question, and takes seriously the question of assessing the relationship between model fit (diagnostic accuracy measured by the AUC) and the practical implications of the work. 1) I thought tying the paper to Swedish policy targets was excellent. As such though, rather than talk about the 5,424 missing cases in the strengths and limitations, the authors could explain in the Study Population section of Population and Methods that the selection criteria are in concordance with those used by the Swedish authorities (as soon as I saw the missing cases at the start of the paper, I was waiting throughout to see how they were addressed – it turns out it was not relevant to the policy question). 2) The paper relies exclusively on AUC as the measure of model fit and derives its policy recommendations from AUC. As such, the authors need to quickly review the strengths and limitations of AUC as a measure of model fit (e.g. that it weights errors of omission and commission equally – that is it does consider whether there are costs to misclassification). I don't know whether AUC is affected by the size of different strata in the analysis, but these sorts of issues should be mentioned. 3) A difficulty the authors face is introducing a novel statistical technique which has been more fully described elsewhere while also addressing a substantive research question, whilst meeting the strict word limits of this journal. That said, I found what AHIDA
--

	is confusing. There is an extensive discussion of AIHDA in the Strengths and Limitations section of the paper. Some of these more general points may not be required. Other aspects (e.g. the use of a fixed effects framework in this paper) need to be introduced earlier in the paper. When I got to the end, I was still unclear on how this paper differed from standard logistic regression. 4) Beginning in the Introduction, the authors make reference to the problem of heterogeneity around averages. But this is not spelled out enough to help the reader. If this needs to be said, it needs to be explained more fully. And if it is not relevant to this paper (as opposed to AIHDA generally) it can be dropped. 5) In the Introduction, the authors assert that for healthcare to be fully efficient it must be equitable. This seems to confuse effectiveness with efficiency. Some sub-populations may be very expensive to offer health services to (e.g. those in remote locations, with cultural needs, or with multiple disadvantage). There might be a strong moral case to service these groups, but probably not on the basis of “efficiency”. 6) In the interests of transparency, the authors might want to consider appending the relevant SAS or Stata code. 7) Minor points:  • In the Strengths and Limitations section of the Abstract, the authors need to choose whether each sentence ends in a full stop or not. • In some places in the text numbers have commas (e.g. n = 29,231) in other places commas have been dropped. This just requires a quick edit for consistency.
--	---

VERSION 1 – AUTHOR RESPONSE

Reviewer: 1

Dr. Lu Yin, Chinese Academy of Medical Sciences and Peking Union Medical College Fuwai Hospital
Comments to the Author:

Even with a large sample size, based on the poor novelty and a potentially wrong design, the study has low power to be published.

AU: It is a pity that the referee does not give any detail as to why they think our approach might use a “potentially wrong design” shows “poor novelty”.

What we can say is that we have carefully considered our study design and we believe it to be correct. Indeed, we have conducted and published a large body of research into multilevel analysis and the analysis of individual heterogeneity and discriminatory accuracy which underpins the current design and we have cited this in the current manuscript. More generally, our research team has a long experience when it comes to developing, applying, and disseminating advanced epidemiological designs and statistical methods for the evaluation of health care.

In terms of the statement of “poor novelty”, we assume this is in reference to the simple nature of our design. This is intentional. Our aim is to illustrate AIHDA as an approach to basic comparisons of healthcare performance which we believe will be accessible to most decision makers without requiring them to have formal epidemiological research experience. We believe that researchers have the duty to inform the wider community and to facilitate the transmission of knowledge from academia to the society and our manuscript is written in this spirit.

We are pleased that the Editor and the other referees see value in the approach we illustrate, and we hope Dr Lu Yin reconsiders their opinion on our manuscript.

Reviewer: 2

Dr. Rossella Miglio, University of Bologna Comments to the Author:

This work is aimed to evaluate both geographical and sociodemographic differences in the statin treatment after AMI in Swedish regions using an AIHDA approach.

The method is simple and can help to understand the effect of the studied variables even if some extension should be considered in my opinion as reported below.

AU: Thank you very much for your comments. We do agree. This was the aim of our work, to provide a simple method that facilitates understanding. Following the recommendations of the referee we provide complementary information as explained below.

The analysis first concentrates on geographical heterogeneity and then on sociodemo-graphic influence on the distribution of the prevalence rate of statin dispensation.

The authors construct multicategorical sociodemographic strata that basically correspond to the interactions between four qualitative sociodemographic variables: age, sex, income and immigration status.

AU: Yes, the multicategorical variable can be used to analyze the interaction of effects between the sociodemographic variables. In previous publications (see e.g., (1)) we have quantified the presence of interaction of effects by comparing the AUC of a model including all the single variables used to define the strata with a model with the multicategorical variable. A positive AUC increment indicates the presence of interaction of effects.

We did not present this information as we aimed to keep the analysis as simple as possible and the focus was on mapping the inequalities rather than on identifying mechanisms. However, following this and other commentaries below we provide supplementary analyses (S1) extending the number of models, which allow us to evaluate the presence of interaction of effects in the multicategorical variable (see below).

Age was considered only using two classes, this could be not effective to evaluate the age effect as reported in the limitation of the study.

AU: We believe that in the case of the quality indicator we analyze, statin use, using two age groups is sufficient as it includes the age of retirement (i.e., 65 years at the time of the study), and it is known that this cut-off is relevant for understand socioeconomic differences in health care utilization (2). However, for other quality indicators it may be better to provide more age classes. We comment this issue in the discussion part of the revised paper.

Moreover, working in this way, it is not possible to evaluate separate effects of each variable and to understand if some of them contribute more to the final results and if there is a real interaction between all this variables.

AU: We agree with the referee. Draft versions of our manuscript included models analyzing the separate effect of each single variable. However, we had in mind to provide a simple, straightforward model that could be used for the evaluation of health care inequalities in routine reporting, rather than to provide a deeper analysis. Therefore, we did not present intermediate analysis as we have done in other publications. Nevertheless, following the opinion of the referee we provide this information as supplementary information (S1). According to this analysis, the AUC is small for all the variables. The highest AUC is for sex (AUC = 0.579) which is 0.029 units lower than the AUC for all variables together (AUC = 0.608). There is a very small interaction of effects in the model with the categorical

variable as, when comparing with the model with all single variables together, the AUC of the model with the multicategorical variable only increases 0.001 units. We now comment on this additional analysis at the end of the Results section.

The model 2 is complex and it is possible that it doesn't predict better than a model with single variables.

AU: We agree, the model including the multicategorical variable does not predict much better than one including the single variables together (Table SM1, Model All). However, the aim of the analysis is to facilitate the understating of the heterogeneity in the distribution of the quality indicator. In general, this information is better captured by modeling the multicategorical variable as it directly maps the distribution of risk across the different strata.

I suggest the authors to consider also alternative models with simple separate effect of each variable even if they are not interested on a mean effect, it is important to verify that the proposed model is effectively better.

AU: We now provide information of such models in the supplementary information (S1) of the revised manuscript (see previous answer). However, the final model including the multicategorical variable is slightly better than a previous model entering the variables separately. This is because the multicategorical variable allows to model interaction of effects. However, the principal aim of our study was neither to identify interaction nor pure prediction but to facilitate the understating of the socioeconomic heterogeneity in the distribution of the quality indicator. We choose a priori variables providing relevant demographic and socioeconomic information.

In the method paragraph they declare to use robust confidence intervals but it is not specified how they were derived.

AU: The robust confidence intervals are heteroskedasticity robust confidence intervals based on the Huber/White/sandwich estimator as implemented via the `vce(robust)` option of the `stcox` command in Stata... By default, `stcox` produces the conventional model-based estimate for the variance-covariance matrix of the coefficients (and hence the reported standard errors). If, however, you specify the `vce(robust)` option, `stcox` switches to the robust variance estimator*. The key to the robust calculation is using the efficient score residual for each subject in the data for the variance calculation. Even in simple single-record, single-failure survival data, the same subjects appear repeatedly in the risk pools, and the robust calculation needs to account for that. A longer explanation is provided in the Stata help files. *Lin, D. Y., and L. J. Wei. 1989. The robust inference for the Cox proportional hazards model. *Journal of the American Statistical Association* 84: 1074–1078.

Moreover it is reported:

"Model 1B estimates sociodemographic inequalities by including the sociodemographic multicategorical variable only, specifying the strata of native 40-64 years old men as the reference category."

I think that something is missing here, we have no information on income (middle income I suppose).

AU: Thank you for this observation. It was a mistake of ours. The correct description is "native 40-64 years old men with high income". We provide the complete definition of the reference group in the revised manuscript

Why it was not considered as an alternative a Poisson regression model (multilevel)?

AU: We think the approach we use is easy to apply, and it is analogous to Poisson regression when follow up time is constant. You could also use logistic regression if the outcome you investigate is rare. See (4) for a longer explanation.

We have experience in the application of multilevel (M) analysis within the AIHDA framework, an alternative that we denominate (M)AIHDA (5) (6-13). But in of our present study we aimed to provide a correct but simpler alternative for routine evaluation in healthcare. See also (1).

Considering the results the discriminatory power is not so high and this is a limit of the model on one side, probably other individual variables could improve the classification performance of the model, on the other side this have positive implication from the public health perspective. It is interesting the interpretation offered in the framework reported in box1.

AU: The key idea in our study is that we are interested in in the classification performance or discriminatory accuracy of just the variables we a priori decided to evaluate. That is, the geo-graphical regions and the demographical and socioeconomic variables. Combining this information with information about the target indicator value, we are able to evaluate geographical and sociodemographic differences in a specific quality indicator. We do so inspired by the concept of proportionate universalism for resource allocation in public health (14) (15)

If the classification performance is low (i.e., small differences) interventions should be universal to the whole population of patients, if the classification performance is high (i.e., large differences) interventions should be “targeted” to the strata with the lowest performance. If the classification performance is middle, interventions should be universal but proportionated to the level of low performance. This is a simple framework for the evaluation of health care performance which we believe provides a better basis for decision making than classical analysis based only on the study of differences between groups averages.

As we commented above, the principal aim of our study was neither to identify interaction nor pure prediction but to facilitate the understating of the socioeconomic and geographical heterogeneity in the distribution of the quality indicator. We choose a priori the variables providing relevant demographic and socioeconomic information. In any case, we agree that more extended information on the social and biomedical characteristics of the patients would improve prediction but, if the aim were prediction, we should have used another approach. (3)

Thank you very much for your constructive criticism.

Reviewer: 3

Mr. Daniel Christensen, Telethon Kids Institute Comments to the Author:

The paper uses a novel statistical approach to assess the question of whether statin dispensation targets are met across regions and sociodemographic strata of Sweden.

This paper addresses a clearly policy-relevant question and takes seriously the question of assessing the relationship between model fit (diagnostic accuracy measured by the AUC) and the practical implications of the work.

AU: Thank you very much for your positive opinion.

1) I thought tying the paper to Swedish policy targets was excellent. As such though, rather than talk about the 5,424 missing cases in the strengths and limitations, the authors could explain in the Study Population section of Population and Methods that the selection criteria are in concordance with those used by the Swedish authorities (as soon as I saw the missing cases at the start of the paper, I was waiting throughout to see how they were addressed – it turns out it was not relevant to the policy question).

AU: We understand, and the referee is right, the selection criteria are in concordance with those used by the Swedish authorities as we study in the manuscript. We do not question the quality indicator, but just present a way of analyzing it. We have now clarified this issue in the manuscript.

2) The paper relies exclusively on AUC as the measure of model fit and derives its policy recommendations from AUC. As such, the authors need to quickly review the strengths and limitations of AUC as a measure of model fit (e.g., that it weights errors of omission and commission equally – that is it does consider whether there are costs to misclassification). I don't know whether AUC is affected by the size of different strata in the analysis, but these sorts of issues should be mentioned.

AU: We have added a paragraph to the Discussion which described these important points. Yes, we agree, if false negatives were more likely to occur than false positives because of information bias, false negatives and false positive should not have the same weight, as assuming the same weight will give misleading AUC values. However, this problem should not be a concern in our study as positive and negative events are easy to identify (a patient uses statins or not) and is not probable that information is differentially biased.

The AUC value may also depend on the size of the group being compared. For instance, a stratum may have a high predicted probability of using statins so all individuals in the stratum are classified as "positive" but if the stratum is very large the number of individuals in the stratum who do not use statins (i.e., false positive) may be large and, therefore, the AUC reduced. This could be prevented by weighting the strata by the inverse of the number of individuals or using other measures of discriminatory accuracy like those based on components of variance.(10) However, in our case the unweighted results are relevant per se. A low unweighted AUC would suggest that any potential intervention to increase use of statins should not be targeted to specific strata with a low predicted probability of statin use but be universal and directed to all the strata since there are many patients in the strata with high predicted probabilities who do not use statins.

3) A difficulty the authors face is introducing a novel statistical technique which has been more fully described elsewhere while also addressing a substantive research question, whilst meeting the strict word limits of this journal. That said, I found what AIHDA is confusing. There is an extensive discussion of AIHDA in the Strengths and Limitations section of the paper. Some of these more general points may not be required. Other aspects (e.g. the use of a fixed effects framework in this paper) need to be introduced earlier in the paper. When I got to the end, I was still unclear on how this paper differed from standard logistic regression.

AU: We need to clarify that our purpose was not to introduce a new statistical technique. Rather, our aim is to introduce an already defined and innovative strategy of analysis. By terming this strategy as "analysis of individual heterogeneity and discriminatory accuracy (AIHDA)" we underline the importance of focusing on components of individual heterogeneity and discriminatory accuracy and not only on differences between group averages (as it is the traditional approach). Within this strategy you can use any statistic technique, like logistic or Poisson regression. We have also developed the AIHDA approach using multilevel models that we denominate MAIHDA (5) (6-13).

Our challenge is double. First, we aim to introduce a strategy of analysis that may discomfort the readers habituated with traditional focus on group mean differences (16). Second, we aim to present this strategy of analysis in a correct but simplified way. The aim of our study is to provide a simple alternative for evaluation of inequalities in routine health care. Still the analytical approach and the way we use some concepts are innovative which may produce some unavoidable initial confusion. In any case, we appreciate the feedback provided by the referee.

The comment of the referee is very relevant as many other readers may have a similar opinion. Therefore, we have commented this issue in the discussion part of the revised manuscript

4) Beginning in the Introduction, the authors make reference to the problem of heterogeneity around averages. But this is not spelled out enough to help the reader. If this needs to be said, it needs to be

explained more fully. And if it is not relevant to this paper (as opposed to AIHDA generally) it can be dropped.

AU: the issue of heterogeneity around averages is still a bit innovative in Epidemiology and healthcare evaluation. However, it is a key issue in (M)AIHDA, and we would prefer to keep it in the revised manuscript. To improve comprehension, we have added some further explanation on this idea in the introduction and discussion part of the revised manuscript.

5) In the Introduction, the authors assert that for healthcare to be fully efficient it must be equitable. This seems to confuse effectiveness with efficiency. Some sub-populations may be very expensive to offer health services to (e.g. those in remote locations, with cultural needs, or with multiple disadvantage). There might be a strong moral case to service these groups, but probably not on the basis of "efficiency".

AU: We understand the idea of the referee. However, in our model of health care quality "efficient" is not only defined by costs. A fundamental dimension of health care quality is "equity". That is, on equal terms and according to needs. Some groups of patients may need more expensive care, and, in this case, it would be "inefficient" to provide cheaper care of worse quality. To be efficient health care needs also be knowledge based, in time, patient centered and safe. All those dimensions need be equitable (i.e., provided on equal terms and according to needs). We believe these ideas are clear in the manuscript. However, now we use the expression "efficient and effective", The term "effectiveness" uses to be contrasted with term "efficacy". While efficacy refers to the effect of an intervention in ideal conditions (e.g., the effect of a medication in a randomized clinical trial), effectiveness indicates the effect of an intervention in the real world setting under routine health care.

6) In the interests of transparency, the authors might want to consider appending the relevant SAS or Stata code.

AU: We have added the Stata codes as supplementary material (S2)

7) Minor points:

- In the Strengths and Limitations section of the Abstract, the authors need to choose whether each sentence ends in a full stop or not.

AU: We have revised this issue.

- In some places in the text numbers have commas (e.g. n = 29,231) in other places commas have been dropped. This just requires a quick edit for consistency.

AU: We have corrected this issue except for calendar years (e.g., 2010 etc)

Thank you very much for your constructive criticism.

REFERENCES

1. Wemrell M, Bennet L, Merlo J. Understanding the complexity of socioeconomic disparities in type 2 diabetes risk: a study of 4.3 million people in Sweden. *BMJ Open Diabetes Res Care*. 2019;7(1):e000749.
2. Merlo J, Gerdtham UG, Lynch J, Beckman A, Norlund A, Lithman T. Social inequalities in health – do they diminish with age? Revisiting the question in Sweden 1999. *Int J Equity Health*. 2003;2(1):2.

3. Moons KG, de Groot JA, Bouwmeester W, Vergouwe Y, Mallett S, Altman DG, et al. Critical appraisal and data extraction for systematic reviews of prediction modelling studies: the CHARMS checklist. *PLoS Med.* 2014;11(10):e1001744.
4. Barros AJ, Hirakata VN. Alternatives for logistic regression in cross-sectional studies: an empirical comparison of models that directly estimate the prevalence ratio. *BMC Med Res Methodol.* 2003;3:21.
5. Merlo J. Multilevel analysis of individual heterogeneity and discriminatory accuracy (MAIHDA) within an intersectional framework. *Soc Sci Med.* 2018;203:74-80.
6. Ljungman H, Wemrell M, Khalaf K, Perez-Vicente R, Leckie G, Merlo J. Antidepressant use in Sweden: an intersectional multilevel analysis of individual heterogeneity and discriminatory accuracy (MAIHDA). *Scand J Public Health.* 2022;50(3):395-403.
7. Zettermark S, Khalaf K, Perez-Vicente R, Leckie G, Mulinari D, Merlo J. Population heterogeneity in associations between hormonal contraception and antidepressant use in Sweden: a prospective cohort study applying intersectional multilevel analysis of individual heterogeneity and discriminatory accuracy (MAIHDA). *BMJ Open.* 2021;11(10):e049553.
8. Rodriguez-Lopez M, Merlo J, Perez-Vicente R, Austin P, Leckie G. Cross-classified Multilevel Analysis of Individual Heterogeneity and Discriminatory Accuracy (MAIHDA) to evaluate hospital performance: the case of hospital differences in patient survival after acute myocardial infarction. *BMJ Open.* 2020;10(10):e036130.
9. Khalaf K, Axelsson Fisk S, Ekberg-Jansson A, Leckie G, Perez-Vicente R, Merlo J. Geographical and sociodemographic differences in discontinuation of medication for Chronic Obstructive Pulmonary Disease - A Cross-Classified Multilevel Analysis of Individual Heterogeneity and Discriminatory Accuracy (MAIHDA). *Clin Epidemiol.* 2020;12:783-96.
10. Merlo J, Wagner P, Leckie G. A simple multilevel approach for analysing geographical inequalities in public health reports: The case of municipality differences in obesity. *Health Place.* 2019;58:102145.
11. Persmark A, Wemrell M, Zettermark S, Leckie G, Subramanian SV, Merlo J. Precision public health: Mapping socioeconomic disparities in opioid dispensations at Swedish pharmacies by Multilevel Analysis of Individual Heterogeneity and Discriminatory Accuracy (MAIHDA). *PLoS One.* 2019;14(8):e0220322.
12. Axelsson Fisk S, Mulinari S, Wemrell M, Leckie G, Perez Vicente R, Merlo J. Chronic Obstructive Pulmonary Disease in Sweden: An intersectional multilevel analysis of individual heterogeneity and discriminatory accuracy. *SSM Popul Health.* 2018;4:334-46.
13. Hernandez-Yumar A, Wemrell M, Abasolo Aleson I, Gonzalez Lopez-Valcarcel B, Leckie G, Merlo J. Socioeconomic differences in body mass index in Spain: An intersectional multilevel analysis of individual heterogeneity and discriminatory accuracy. *PLoS One.* 2018;13(12):e0208624.
14. Marmot M, Bell R. Fair society, healthy lives. *Public Health.* 2012;126 Suppl 1:S4-S10.
15. Carey G, Crammond B, De Leeuw E. Towards health equity: a framework for the application of proportionate universalism. *Int J Equity Health.* 2015;14:81.
16. Merlo J. Invited commentary: multilevel analysis of individual heterogeneity-a fundamental critique of the current probabilistic risk factor epidemiology. *Am J Epidemiol.* 2014;180(2):208-12; discussion 13-4.

VERSION 2 – REVIEW

REVIEWER	Christensen, Daniel Telethon Kids Institute
REVIEW RETURNED	16-Jan-2023
GENERAL COMMENTS	I'll start with the general comment that saying "we made changes" and sending back a track changes document without pointing the reviewer to specific points in the paper where changes have been

	made moves the burden from the author to the reviewer in a way that I'm unaccustomed to. It looks like some previous comments have not been addressed. While I still maintain the paper has worthwhile content, considerable effort may be required to make the paper readable enough for this journal. #1 I thought tying the paper to Swedish policy targets was excellent. As such though, rather than talk about the 5,424 missing cases in the strengths and limitations, the authors could explain in the Study Population section of Population and Methods that the selection criteria are in concordance with those used by the Swedish authorities (as soon as I saw the missing cases at the start of the paper, I was waiting throughout to see how they were addressed – it turns out it was not relevant to the policy question). AU: We understand, and the referee is right, the selection criteria are in concordance with those used by the Swedish authorities as we study in the manuscript. We do not question the quality indicator, but just present a way of analyzing it. We have now clarified this issue in the manuscript. >>> I can't find the change. Please flag where it is. #3 A difficulty the authors face is introducing a novel statistical technique which has been more fully described elsewhere while also addressing a substantive research question, whilst meeting the strict word limits of this journal. That said, I found what AIHDA is confusing. There is an extensive discussion of AIHDA in the Strengths and Limitations section of the paper. Some of these more general points may not be required. Other aspects (e.g. the use of a fixed effects framework in this paper) need to be introduced earlier in the paper. When I got to the end, I was still unclear on how this paper differed from standard logistic regression. AU: We need to clarify that our purpose was not to introduce a new statistical technique. Rather, our aim is to introduce an already defined and innovative strategy of analysis... >>> I appreciate the challenge the authors face. But I maintain that more work needs to be done in the Introduction and Methods to explain concepts, rather than introducing these in the Discussion. Putting a paragraph at the very end of the paper saying "We need to clarify that our purpose was not to introduce a new statistical technique. Rather, our aim is to introduce an innovative strategy of analysis..." doesn't help the reader orient themselves to the paper. Nor does putting in a note at the end "sorry if this is tough on the reader". Instead, some more work needs to be done at the front of the paper introducing concepts. The paragraph in the Introduction on heterogeneity around averages could be improved by an illustrative example. While the Introduction talks about AIHDA's benefits, it doesn't say what it is. I still don't see how the MAIHDA paragraph progresses the paper. Additionally, the only change to this paragraph was to put the word "conceptually" at the start. Per my previous review, this is also the first time in the paper level 1 and level 2 are discussed. If the concepts are worthy enough for the Discussion, they need to be set up earlier in the paper.
--	--

	#4 Beginning in the Introduction, the authors make reference to the problem of heterogeneity around averages. But this is not spelled out enough to help the reader. If this needs to be said, it needs to be explained more fully. And if it is not relevant to this paper (as opposed to AIHDA generally) it can be dropped. AU: the issue of heterogeneity around averages is still a bit innovative in Epidemiology and healthcare evaluation. However, it is a key issue in (M)AIHDA, and we would prefer to keep it in the revised manuscript. To improve comprehension, we have added some further explanation on this idea in the introduction and discussion part of the revised manuscript. >>> I cannot find these changes.
--	--

VERSION 2 – AUTHOR RESPONSE

Reviewer: 3

Mr. Daniel Christensen, Telethon Kids Institute Comments to the Author:

I'll start with the general comment that saying, "we made changes" and sending back a track changes document without pointing the reviewer to specific points in the paper where changes have been made moves the burden from the author to the reviewer in a way that I'm unaccustomed to.

It looks like some previous comments have not been addressed. While I still maintain the paper has worthwhile content, considerable effort may be required to make the paper readable enough for this journal.

AU: Thank you very much for your appreciation of our work. We are sorry we have not clearly indicated where the changes were done in the revised manuscript. It is very important that the paper is readable and in our second revision we have try our best to improve this issue. We follow BMJ Open guidelines and provide a clean version and a version with marked changes.

#1 I thought tying the paper to Swedish policy targets was excellent. As such though, rather than talk about the 5,424 missing cases in the strengths and limitations, the authors could explain in the Study Population section of Population and Methods that the selection criteria are in concordance with those used by the Swedish authorities (as soon as I saw the missing cases at the start of the paper, I was waiting throughout to see how they were addressed – it turns out it was not relevant to the policy question).

AU: We understand, and the referee is right, the selection criteria are in concordance with those used by the Swedish authorities as we study in the manuscript. We do not question the quality indicator, but just present a way of analysing it. We have now clarified this issue in the manuscript.

>>> I can't find the change. Please flag where it is.

#1 AU: Following the advice of the referee, we have now included this information in the section of "Study population". However, this information was already in the discussion section of the manuscript in the discussion: "Moreover, we can directly compare our results to those presented by the NBHW since the definition of the quality indicator is the same." And also: "One limitation of this study is that we excluded the 5,424 individuals who died, emigrated, or had missing information on country of birth

which corresponds to approximately 13% of the final study population, and it cannot be ruled out that this exclusion introduces a selection bias. However, we do not think that this limitation alters our conclusions as the selection criteria are in concordance with the those used by the Swedish authorities.”

#3 A difficulty the authors face is introducing a novel statistical technique which has been more fully described elsewhere while also addressing a substantive research question, whilst meeting the strict word limits of this journal. That said, I found what AHIDA is confusing. There is an extensive discussion of AIHDA in the Strengths and Limitations section of the paper. Some of these more general points may not be required. Other aspects (e.g. the use of a fixed effects framework in this paper) need to be introduced earlier in the paper. When I got to the end, I was still unclear on how this paper differed from standard logistic regression.

AU: We need to clarify that our purpose was not to introduce a new statistical technique. Rather, our aim is to introduce an already defined and innovative strategy of analysis...

>>> I appreciate the challenge the authors face. But I maintain that more work needs to be done in the Introduction and Methods to explain concepts, rather than introducing these in the Discussion.

Putting a paragraph at the very end of the paper saying “We need to clarify that our purpose was not to introduce a new statistical technique. Rather, our aim is to introduce an innovative strategy of analysis...” doesn’t help the reader orient themselves to the paper. Nor does putting in a note at the end “sorry if this is tough on the reader”. Instead, some more work needs to be done at the front of the paper introducing concepts.

The paragraph in the Introduction on heterogeneity around averages could be improved by an illustrative example

While the Introduction talks about AIHDA’s benefits, it doesn’t say what it is.

I still don’t see how the MAIHDA paragraph progresses the paper.

Additionally, the only change to this paragraph was to put the word “conceptually” at the start. Per my previous review, this is also the first time in the paper level 1 and level 2 are discussed. If the concepts are worthy enough for the Discussion, they need to be set up earlier in the paper.

AU #3: Thank you for your advice. We do try to introduce the AIHDA approach as accessible as possible for most readers. Following the comments of the referee we have made some text addition and reorganized the the manuscript.

1.- We have moved fragments of the explanations in the discussion part to the introduction part. The moved text is possible to identify in the Word version showing the changes.

2.- Concerning the issue on heterogeneity around averages, we have published a full paper aimed to introduce the concept. We think the most appropriate is to strongly suggest the readers to examine this previous publication: Merlo J, Wagner P, Leckie G. A simple multilevel approach for analysing geographical inequalities in public health reports: The case of municipality differences in obesity. Health Place. We have included an explanation and recommendations on further reading.

3.- By doing the changes described above we have also introduced the level 1 and level 2 concepts in the introduction section of the revised manuscript (see page 5, 2nd paragraph)

#4 Beginning in the Introduction, the authors make reference to the problem of heterogeneity around averages. But this is not spelled out enough to help the reader. If this needs to be said, it needs to be explained more fully. And if it is not relevant to this paper (as opposed to AIHDA generally) it can be dropped.

AU: the issue of heterogeneity around averages is still a bit innovative in Epidemiology and healthcare evaluation. However, it is a key issue in (M)AIHDA, and we would prefer to keep it in the revised manuscript. To improve comprehension, we have added some further explanation on this idea in the introduction and discussion part of the revised manuscript.

>>> I cannot find these changes.

AU #4: We believe this issue is better confronted in the revised manuscript See our comments (AU #3.) above.

VERSION 3 – REVIEW

REVIEWER	Christensen, Daniel Telethon Kids Institute
REVIEW RETURNED	14-Apr-2023

GENERAL COMMENTS	Thanks to the authors for their response. A few minor comments #1. The missing data. > Personally, given what you are doing concords with the official definitions, I think it's a strength not a limitation, but this is adequately addressed as is. #2. Introduction of statistical concepts. > Heterogeneity The paragraph on page 4 on heterogeneity needs a "because..." or "for example...", sentence to finish the paragraph. > (M)AIHDA The paragraph on page 6 beginning "AIHDA can be applied using traditional regression..." claims that MLRA uses grand mean centering. Maybe I'm mistaken, but I thought different approaches to mean centering was an active topic in the MRLA literature? Hopefully the relevant sentence can be reworded to make this clear (or deleted if it doesn't build to the Discussion).
--

VERSION 3 – AUTHOR RESPONSE

Reviewer: 3

Mr. Daniel Christensen, Telethon Kids Institute Comments to the Author:

Thanks to the authors for their response. A few minor comments

AU: Thank you for your comments. We have made changes accordingly.

#1. The missing data. > Personally, given what you are doing concurs with the official definitions, I think it's a strength not a limitation, but this is adequately addressed as is.

AU: We agree that using an official and standardized definition of the quality indicator is a strength. The use of national registers with valid and standardized information on diagnoses, medication use and sociodemographic variables is also considered to be a strength. Our missing data (5,424 individuals, approximately 13% of the study population) could however introduce a selection bias, which is part of our discussion.

#2. Introduction of statistical concepts.

> Heterogeneity The paragraph on page 4 on heterogeneity needs a "because..." or "for example...", sentence to finish the paragraph.

AU: This issue has been addressed with an added example in the mentioned paragraph.

> (M)AIHDA The paragraph on page 6 beginning "AIHDA can be applied using traditional regression..." claims that MLRA uses grand mean centering. Maybe I'm mistaken, but I thought different approaches to mean centering was an active topic in the MRLA literature? Hopefully the relevant sentence can be reworded to make this clear (or deleted if it doesn't build to the Discussion).

AU: In our opinion, both approaches provided relevant information and need be interpreted correctly. As explained in our paper, our aim was to provide an accessible analytical approach that can be used and easily interpreted in routine evaluation of health care quality. The interested reader may obtain a deeper understanding on this issue elsewhere. 25 26 27.

Reviewer: 3 Competing interests of Reviewer: N/A